# Low prevalence of human pegivirus 1 (HPgV-1) in HTLV-1 carriers from Belém, Pará, North Region of Brazil

**Ana Beatriz Figueiredo de Lima**[1], **Keyla Santos Guedes de Sá**[1], **Maria Karoliny da Silva Torres**[1], **Luana da Silva Soares**[2], **Hugo Reis Resques**[2], **Vânia Nakauth Azevedo**[3], **Rosimar Neris Martins Feitosa**[3], **Jacqueline Cortinhas Monteiro**[3], **Andrea Nazaré Monteiro Rangel da Silva**[3], **Andre Luis Ribeiro Ribeiro**[3], **Aldemir Branco de Oliveira-Filho**[4], **Antonio Carlos Rosario Vallinoto**[3], **Luiz Fernando Almeida Machado**[1,3¤] *

1 Programa de Pós-graduação em Biologia de Agentes Infecciosos e Parasitários, Instituto de Ciências Biológicas, Universidade Federal do Pará, Belém, Pará, Brazil, 2 Laboratório de Virologia, Serviço de Vigilância em Saúde, Ministério da Saúde, Instituto Evandro Chagas, Ananindeua, Pará, Brazil, 3 Laboratório de Virologia, Instituto de Ciências Biológicas, Universidade Federal do Pará, Belém, Pará, Brazil, 4 Grupo de Estudo e Pesquisa em Populações Vulneráveis, Instituto de Estudos Costeiros, Universidade Federal do Pará, Bragança, Pará, Brazil

¤ Current address: Laboratório de Virologia, Instituto de Ciências Biológicas, Universidade Federal do Pará, Belém, Pará, Brazil
* lfam@ufpa.br

**Data Availability Statement:** All relevant data are within the manuscript.

## Abstract

### Introduction

Human pegivirus 1 (HPgV-1) is a single-stranded, positive-sense RNA virus belonging to the *Flaviviridae* family with limited cause-effect evidence of the causation of human diseases. However, studies have shown a potential beneficial impact of HPgV-1 coinfection in HIV disease progression. Human T lymphotropic virus-1 (HTLV-1) is a retrovirus known for causing diseases, especially in muscle and white blood cells, in approximately 5% of patients. Thus, this study aimed to investigate the potential effects of an HPgV-1 infection in patients carrying HTLV-1 in the state of Pará in the North Region of Brazil.

### Methods

A group of HTLV-1 carriers was compared to healthy controls. Blood samples were collected, data from medical regards were collected, and a questionnaire was administered. HPgV-1 and HTLV-1 positivity was determined by quantitative polymerase chain reaction (qRT-PCR). The data were analyzed to correlate the effects of HPgV-1 coinfection in HTLV-1 carriers.

### Results

A total of 158 samples were included in the study: 74 HTLV-1-positive patients (46,8%) and 84 healthy controls (53,2%). The overall HPgV-1 positivity rate was 7.6% (12/158), resulting in a prevalence of 5.4% (4/74) and 9.5% (8/84) in HTLV-1 carriers and healthy controls,

**Funding:** This study was financed in part by the Coordenação de Aperfeiçoamento de Pessoal de Nível Superior - Brasil (CAPES) - Finance Code 001. The funders had no role in study design, data collection and analysis, decision to publish, or preparation of the manuscript.

**Competing interests:** The authors have declared that no competing interests exist.

respectively. No significant differences were found when comparing any clinical or demographic data between groups.

## Conclusion

This study indicated that the prevalence of HPgV-1 infection is low in HTLV-1 carriers in Belém, Pará, and probably does not alter the clinical course of HTLV-1 infection, however, further studies are still needed.

## Introduction

Human pegivirus 1 (HPgV-1) was discovered in 1995 and is thought to be an etiological agent for non-A to E hepatitis. However, well-controlled, prospective studies failed to identify an association between an infection and acute or chronic hepatitis [1]. HPgV-1 was formerly known as hepatitis G virus/GB virus C (GBV-C) and is a single-stranded, positive-sense RNA virus belonging to the *Flaviviridae* family and the *Pegivirus* genus [2]. Although it has a high prevalence (studies suggest that there are ~ 750 million people with HPgV-1 infection worldwide), there is limited evidence for HPgV-1 as a primary etiological factor in human diseases [3]. On the other hand, HPgV-1 infection has been linked to modulating the course of other viral diseases, including human immunodeficiency virus (HIV) infection/acquired immunodeficiency syndrome (AIDS), with a supposed beneficial effect; however, little is known about HPgV-1 coinfection in other viral diseases [4,5].

HPgV-1 is transmitted by exposure to infected blood, mainly through sexual exposure or by maternal–fetal transmission; cross-sectional serum surveys indicate between 1–5% of HPgV-1 viremia cases occur in developed countries, while up to 20% of blood donors in developing countries have an active infection [6,7]. Studies suggest a positive effect of chronic HPgV-1 infection in HIV-infected patients, in which data show a higher $CD4^+$ T-cell count, lower HIV viral load and inflammatory markers, and delayed progression to AIDS [8,9]. HIV infection results in chronic activation of T cells, promoting activation-induced $CD4^+$ T-cell death, resulting in lower $CD4^+$ T-cell counts and progression to AIDS [10]. Conversely, HPgV-1 infection is associated with the reduced activation of T-cells in HIV-infected individuals compared to those without HPgV-1, which can help in the long life of those infected with HIV-1 [7,11,12]. Still intriguing, a study also indicated that HPgV-1 can interact with the host's immune system and modulate the super-exuberant immune response of the pathogenesis related to Ebola virus (EBOV) infection [13]. Thus, considerable attention has been given to investigating the association between the HPgV-1/HIV coinfection and how it can potentially improve the outcomes in HIV-infected individuals; however, limited data are found between the association and effects of HPgV-1 coinfection with the rare human T lymphotropic virus-1 (HTLV-1) infection [14].

HTLV-1 was the first human retrovirus and was discovered in 1980; HTLV-1 is found in diverse regions of the world, where its prevalence is estimated to infect approximately 10 to 20 million people worldwide [15]. Although most HTLV-1 carriers remain asymptomatic, approximately 5% of infected individuals can develop clinical manifestations, including adult T-cell leukemia/lymphoma (ATL) and tropical spastic paraparesis/HTLV-1-associated myelopathy (TSP/HAM) [16]. Furthermore, other inflammatory manifestations, including uveitis, dermatitis and rheumatological disorders, have also been associated with HTLV-1 infection [17,18].

In Brazil, the seroprevalence of viral infections is quite diverse. HTLV-1 seroprevalence in all 27 state capital cities varied from 0.04% to 1% in healthy blood donors. Other studies have also investigated the prevalence of HPgV-1 in healthy blood donors in the South Region, and the results indicated a seroprevalence between 5.9% [6] and 21.7% [19]. In people living with HIV, the prevalence of HPgV-1 infection was 34% [20], and vertical transmission of HPgV-1 in pregnant women with HIV was 31% [21].

The epidemiology of HPgV-1 infection in northern Brazil is still unknown, as there are no data about HPgV-1 and HTLV-1 coinfection. Thus, this study aimed to investigate the prevalence of HPgV-1 infection in HTLV-1 carriers with different clinical manifestations and compare them to a control group of HTLV-negative health volunteers in the city of Belém, Pará, North region of Brazil.

## Materials and methods

### Type of study and ethical aspects

This is a descriptive, cross-sectional and observational study. The study was divided into two groups based on the HTLV-1 status (positive or negative). HTLV-1 and HPgV-1 infections were determined by blood sampling. The participants were informed about the objectives of this investigation, and those who agreed to participate in this research signed a consent form. Demographic data on age, sex, and clinical symptoms were obtained from medical records. This study was approved by the Human Research Ethics Committee of the Institute of Health Sciences of the Federal University of Pará under protocol number 2.305.226.

### Study population

The study group included 74 HTLV-1 carriers (18 TSP/HAM symptomatic patients, 3 with rheumatologic symptoms, 1 with uveitis, 53 asymptomatic and 4 with no clinical information) attending the outpatient clinic of the Nucleus of Tropical Medicine of the Federal University of Pará and 84 seronegative healthy volunteers blood donors from the Fundação Centro de Hemoterapia e Hematologia do Pará (HEMOPA) that were stored at the Virology Laboratory of the Federal University of Pará. All individuals in the control group were recruited between June 2012 and September 2012, and individuals from the HTLV-1 carrier group were recruited between 2013 and 2016. The HTLV-1 carriers with other known viral infectious diseases (selected in the medical record) were excluded from this study. All plasma samples from HTLV-1 carriers were confirmed using an immunenzymatic assay for the presence of antibodies anti-HTLV (HTLV-1/2 Ab-Capture ELISA Test System, Ortho Diagnostic Systems Inc., USA) according to the manufacturer's protocol and HTLV infection was confirmed by RFLP [22].

### Sampling

Blood samples (5 mL) were collected from each participant and placed in a tube containing ethylenediaminetetraacetic acid (EDTA) as an anticoagulant. Samples were transferred and stored in the Virology Laboratory of the Federal University of Pará. Plasma and red blood cells were separated by centrifugation at 2,500 rpm for 10 minutes, transferred to Eppendorf tubes and stored at -20˚C prior to molecular biology analyses.

### RNA extraction

Total RNA was extracted from 500 μL of plasma with the Abbott mSample Preparation System (Promega Corporations) following the manufacturer's specifications. RNA was eluted at a final volume of 25 μL of elution buffer and stored at -70˚C.

### HPgV-1 detection

HPgV-1 RNA was detected by nested RT-PCR amplification of the 5' untranslated region (UTR) as described previously [23] with small modifications. In summary, 5 μL of RNA was added to 17 μL of a master mix composed of 5.6 μL of RNase-free $H_2O$, 10 μL of (2x) TaqMan RT PCR Mix, 0.5 μL of (40x) TaqMan RT Enzyme Mix, 0.3 μL of Primer RTG1F (20 nM) 5′ – GTGGTGGATGGGTGATGACA–3′, 0.3 μL of Primer RTG2R (20 nM) 5′–GACCCACCTATAG TGGCTACCA–3′, and 0.3 μL of Probe NFQ (10 nm) 5′–FAM–CCGGGATTTACGACCTACC– 3′.

All samples were tested in duplicate and placed in 96-well microplates, sealed with adhesives and read on an ABI Prism 7500 (Applied Biosystems, Foster City, CA). Reaction cycles were established as follows: 30 minutes at 50˚C (reverse transcription), 10 minutes at 95˚C (denaturation) and 40 cycles of 15 seconds at 95˚C and 60 seconds at 60˚C (annealing and extension). The maximum threshold cycle (Ct) established for positivity was 37, and those above the proposed value were submitted to a new amplification to confirm the results.

### Statistical analysis

For descriptive analyses, the means and standard deviations are reported. A chi-squared test was performed to analyze data in contingency tables. Statistical analyses were carried out using GraphPad Prism 5.0 (GraphPad Software, Inc., San Diego, CA). Differences were considered statistically significant when $p < 0.05$.

## Results

A total of 158 individuals were enrolled in this study, with 74 (46.8%) belonging to the HTLV-1 group and 84 (53.2%) belonging to the control (healthy) group (HTLV-1-negative). The general average age was 44 years (SD = ±13). The mean age of HTLV-1 carriers was 50 years (SD = ±16); most were female (55/74; 74.3%), over 50 years old (37/74; 50%) and asymptomatic (48/74; 64.9%). The control group showed an average age of 39 years (SD = ±7.8); most were female (44/84; 52.4%) and between the ages of 31 and 50 years (69/84; 82.1%).

The overall prevalence of HPgV-1 was 7.6% (12/158). The prevalence of HPgV-1 in HTLV-1 carriers and the control group was 5.4% (4/74) and 9.5% (8/84), respectively. No significant differences were found when comparing the prevalence of HPgV-1-positive individuals between the HTLV-1 and control groups (p = 0.4974) (Table 1). Likewise, no statistically significant differences were observed among the epidemiological and clinical variables and the prevalence of HPgV-1 between the evaluated groups (Table 2).

In HLTV-1-infected individuals, HPgV-1 coinfection was detected only in those above 31 years old. As a result, it was not possible to assess the relationship between age and the two groups of HLTV-1-infected individuals. On the other hand, no differences were found in HPgV-1 infection with regard to sex or symptomatic status of HTLV-1 disease status (Table 2).

## Discussion

This was the first study about the prevalence of HPgV-1 in HTLV-1 carriers in the North Region of Brazil. The prevalence of HPgV-1 infection in HTLV-1 carriers from Pará was 7.6%. This prevalence was much lower than that observed in individuals with HIV in the same region (North Region, 17.0%) [24] and in the South Region of Brazil (21.7%) [20] as well as in HIV carriers from China (23.4%) [7] and Indonesia (88.8%) [11]. We observed a prevalence of

**Table 1. HPgV-1 prevalence in individuals with or without HTLV-1 infection.**

| HPgV-1 | HTLV-1 + | | Health controls | | Total | | p-value |
|---|---|---|---|---|---|---|---|
| | n | % | n | % | n | % | |
| Positive | 4 | 2.5 | 8 | 5.1 | 12 | 7.6 | 0.38[a] |
| Negative | 70 | 44.3 | 76 | 48.1 | 146 | 92.4 | |

HPgV-1, Human Pegivirus 1; HTLV-1, Human T Lymphotropic Virus-1

[a] chi-squared test

HPgV-1 similar to that observed in blood donors from several other Brazilian regions, where rates range from 7.1% to 10.0% [25,26].

On the other hand, the prevalence of HPgV-1 in HTLV-1 carriers in this study was higher than that reported in Nagasaki, Japan (2.8%) [14]. Although HPgV-1 infected some individuals with HTLV-1 in our sample, no differences were observed between the HTLV-1-positive and control groups. This suggests that HTLV-1 does not interfere with the susceptibility to HPgV-1 infection.

This reduced prevalence of HPgV-1 infection compared to most reported papers in the literature may be explained by the method used (only viral RNA was used for diagnosis) [7,14,21,26] and a more advanced age. Individuals between 18 to 30 and 31 to 50 years are approximately eight and six times, respectively more prone to show HPgV infection than those older than 50 years [19].

The prevalence of HPgV-1 is known to be higher in individuals with other viral diseases, especially HIV [19,27]. However, our results showed that it might not occur in patients with HTLV-1 infection. No differences were found between the prevalence of HTLV-1 carriers and healthy controls. Studies suggest that in immunocompetent individuals, the clearance of HPgV-1 occurs during the first years of infection, and HPgV-1 clearance is delayed in

**Table 2. Demographic and disease characteristics between HTLV-1 infection only and HTLV-1 and HPgV-1 coinfection.**

| Demographic and clinical data | HTLV-1 + HPgV-1 | | HTLV-1 only | | Total | | p-value[a] |
|---|---|---|---|---|---|---|---|
| | n | % | n | % | N | % | |
| **Age (years)** | | | | | | | |
| 1–17 | 0 | 0 | 3 | 4.3 | 3 | 4 | NP |
| 18–30 | 0 | 0 | 4 | 5.7 | 4 | 5.4 | |
| 31–50 | 3 | 75 | 20 | 25.6 | 23 | 31.1 | |
| >50 | 1 | 25 | 36 | 51.4 | 37 | 50 | |
| Not available | 0 | 0 | 7 | 10 | 7 | 9.5 | |
| **Sex** | | | | | | | |
| Male | 2 | 50 | 17 | 24.3 | 19 | 25.7 | 0.27 |
| Female | 2 | 50 | 53 | 75.7 | 55 | 74.3 | |
| **Presence of HTLV-1 symptoms** | | | | | | | |
| Asymptomatic | 2 | 50 | 46 | 65.7 | 48 | 64.9 | 0.58 |
| Symptomatic | 2 | 50 | 20 | 28.6 | 22 | 29.7 | |
| Not available | - | - | 4 | 5.7 | 4 | 5.4 | |

HTLV-1, Human T Lymphotropic Virus-1; HPgV-1, Human Pegivirus 1; NP, not performed

[a] chi-squared test, comparing HTLV-1 + HPgV-1 and HTLV-1 only groups

individuals with HIV due to their immunocompromised defenses. Thus, a delayed viral clearance associated with a similar method of transmission was indicated by some authors as a contributing factor for a higher prevalence of HPgV-1 in people living with HIV/AIDS [28–30].

We observed a limited-to-no effect in HTLV-1 carriers when coinfected with HPgV-1, while HPgV-1 coinfection is suggested to delay disease progression in HIV-positive patients. HPgV-1 seems to antagonize some mechanisms involved in the pathophysiology of HIV, including HIV-mediated activation of T-cells and T-cell death [10]. On the other hand, the two most common diseases associated with HTLV-1, ATL and TSP/HAM, have little to benefit from any similar effects. Indeed, ATL is linked to a suppression of the cytotoxic T-lymphocyte function, and TSP/HAM is linked to an elevated cellular acquired immune response and high production of proinflammatory cytokines. Although other common mechanisms could be involved, it is unlikely that HPgV infection would interfere with HTLV-1-related diseases [31].

This study has limitations and should be considered. The small sample size makes it difficult to assess and, consequently, the indication or safe exclusion of possible associations. Another fact to be considered is the instability of the RNA molecule that can be degraded more easily over time, so the time between the collection and the evaluation of the presence of RNA in the samples of participants with low viral load may have caused a false result negative. However, this study presented relevant information on HPgV-1 infection and is also the first investigation of its kind in this remote region of Brazil. Further investigations are necessary to strengthen the data gathered herein, and it seems especially important to prospectively investigate these individuals to establish the effects of HPgV-1 and HTLV-1 coinfection.

In conclusion, the prevalence of HPgV-1 infection in patients with HTLV-1 in the city of Belém is relatively similar to the prevalence found in healthy individuals (seronegative HTLV-1), however this may be influenced by the small sample size. Thus, further studies are needed to confirm that HTLV-1 and HPgV-1 coinfection does not affect the course or development of symptoms related to HTLV-1 infection in affected individuals, since the pathological mechanisms of these viruses are quite different.

## Acknowledgments

The authors thank all the individuals who took part in the study, CAPES (Coordenação de Aperfeiçoamento de Pessoal de Nivel Superior), and the Federal University of Pará.

## Author Contributions

**Conceptualization:** Ana Beatriz Figueiredo de Lima, Antonio Carlos Rosario Vallinoto, Luiz Fernando Almeida Machado.

**Data curation:** Keyla Santos Guedes de Sá, Maria Karoliny da Silva Torres, Luana da Silva Soares, Hugo Reis Resques, Vânia Nakauth Azevedo, Rosimar Neris Martins Feitosa.

**Formal analysis:** Ana Beatriz Figueiredo de Lima, Jacqueline Cortinhas Monteiro, Aldemir Branco de Oliveira-Filho, Luiz Fernando Almeida Machado.

**Funding acquisition:** Luiz Fernando Almeida Machado.

**Methodology:** Ana Beatriz Figueiredo de Lima, Luana da Silva Soares, Hugo Reis Resques, Andrea Nazaré Monteiro Rangel da Silva, Andre Luis Ribeiro Ribeiro.

**Supervision:** Luiz Fernando Almeida Machado.

**Writing – original draft:** Ana Beatriz Figueiredo de Lima, Antonio Carlos Rosario Vallinoto.

**Writing – review & editing:** Ana Beatriz Figueiredo de Lima, Aldemir Branco de Oliveira-Filho, Luiz Fernando Almeida Machado.

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
