## [Decision Letter · Decision Letter 0]

3 Mar 2020

PONE-D-20-02565

Low prevalence of human pegivirus (HPgV) in HTLV-1 carriers from Belém, Pará, Amazon Region of Brazil

PLOS ONE

Dear Dr. Machado,

Thank you for submitting your manuscript to PLOS ONE. After careful consideration, we feel that it has merit but does not fully meet PLOS ONE’s publication criteria as it currently stands. Therefore, we invite you to submit a revised version of the manuscript that addresses the points raised during the review process.

We would appreciate receiving your revised manuscript by Apr 17 2020 11:59PM. To enhance the reproducibility of your results, we recommend that if applicable you deposit your laboratory protocols in protocols.io, where a protocol can be assigned its own identifier (DOI) such that it can be cited independently in the future. For instructions see: http://journals.plos.org/plosone/s/submission-guidelines#loc-laboratory-protocols

We look forward to receiving your revised manuscript.

Kind regards,

Jason Blackard, PhD

Academic Editor

PLOS ONE

Additional Editor Comments (if provided):

This is a cross-sectional study of human pegivirus in HTLV-1 carriers in Brazil.  Given the beneficial effects of HPgV on HIV disease, evaluating the presence/absence of a similar association with HTLV-1 is important.

The overall sample size is moderate (n = 158), and the prevalence of HPgV was 7.6% (a bit lower than for HIV/HPgV co-infection rates in most settings).

Sample storage at -20C may have contributed to degradation of viral RNA.

How big is the HPgV PCR amplicon that was generated?

Were HPgV viral loads or HPgV genotypes evaluated?

Journal Requirements:

2. Please amend either the title on the online submission form (via Edit Submission) or the title in the manuscript so that they are identical.

Reviewers' comments:

Reviewer's Responses to Questions

**Comments to the Author**

1. Is the manuscript technically sound, and do the data support the conclusions?

Reviewer #1: Partly

Reviewer #2: Partly

2. Has the statistical analysis been performed appropriately and rigorously? 

Reviewer #1: Yes

Reviewer #2: Yes

3. Have the authors made all data underlying the findings in their manuscript fully available?

Reviewer #1: Yes

Reviewer #2: Yes

4. Is the manuscript presented in an intelligible fashion and written in standard English?

Reviewer #1: Yes

Reviewer #2: No

5. Review Comments to the Author

Reviewer #1: This paper evaluates the prevalence of HPgV viremia in HTLV+ patients and healthy controls in a region in Brazil. Several issues need to be clarified and the conclusions are not fully supported by the data presented.

1. Please define what is meant by HTLV +. I assume this means seropositive but this isnt clear. Also, what assay was used to determine seropositivity?

2. The data do not support conclusions about susceptibiltity. Susceptibility to viral infections is mediated by many factors and this small study with low numbers of HPgV+ cases measured at one time point retrospectively cannot provide any data about susceptibility.

3. The age difference between the HTLV+ group and the control group is an issue, as younger subjects may be more likely to harbor HPgV as they have had less time to clear it. An age matched control group may show different results.

4. While the study shows no difference in clinical outcomes of HTLV based on HPgV status, the numbers are too small to make any conclusions about this. However no obvious safety signal is seen.

5. Grammatical issues with sentence on lines 63-65 under Introduction.

Reviewer #2: This paper describes HPgV-1 prevalence in people with (n=74) and without (n=84) HTLV-I infection living in Northern Brazil. The prevalence was 5.4% in those with HTLV-I (n= 4), and higher (9.5%) in those without HTLV-I (n=8). Due to the small number of HPgV-1 infections, these differences were not statistically significant. The following represent a number of suggestions that would strengthen the paper.

Abstract:

1. A second human pegivirus (HPgV-2) has been identified, so the HPgV described here is best called HPgV-1.

2. The conclusion that HTLV-1 “does not change the susceptibility of patients to acquiring an HPgV infection” should be qualified, as the statistical power to suggest an increase or decrease in infection given the numbers is very low.

Introduction

3. In addition to surrogate markers of HIV-1 disease progression mentioned in lines 75 – 79, the association between HPgV-1 and survival in HIV-1 infected people is not mentioned. The data are extensive showing reduced mortality in HIV-1 patients, even in incident infection (Vahidnia et al, JID, 2012). Also, survival in Ebola-infected people was higher in those with HPgV-1 infecton (Lauck et al., J Virol, 2014). This information should be noted.

Methods: No comments

Results:

4. The question of HPgV-1 prevalence and age differentiation is not possible to address due to the small number of individuals studied in age groups < 30. It appears there were 7 in the HTLV-1-positive group, and it is not clear how many individuals were in this age group in the HPgV-1+HTLV-1 coinfection group (only data for the 4 HPgV-1 cases were provided, no denominator).

In conclusion, this is the second study to examine HPgV-1 infection in HTLV-I infected adults, and the first from this geographical location. Although the data do not identify significant differences in HPgV-1 prevalence by HTLV-1 status, age, sex, etc., the power to detect differences is limited due to the small number of HPgV-1 infections (n=12/158). There are many places throughout the manuscript where attention to English usage should be provided.

6. PLOS authors have the option to publish the peer review history of their article (what does this mean?). If published, this will include your full peer review and any attached files.

Reviewer #1: No

Reviewer #2: No

---

## [Author Response · Author response to Decision Letter 0]

14 Apr 2020

Reviewer #1: This paper evaluates the prevalence of HPgV viremia in HTLV+ patients and healthy controls in a region in Brazil. Several issues need to be clarified and the conclusions are not fully supported by the data presented.

1. Please define what is meant by HTLV+. I assume this means seropositive but this isnt clear. Also, what assay was used to determine seropositivity?

RESPONSE: We are grateful for the observation, HTLV+ individuals are those who were positive on the serological test (ELISA) and were later confirmed by PCR-RFLP. This information was added to the "Study population" as shown below (lines 132-136): 

“All plasma samples dos indivíduos com HTLV were confirmados using an immunenzymatic assay for the presence of antibodies anti-HTLV (HTLV-1/2 Ab-Capture ELISA Test System, Ortho Diagnostic Systems Inc., USA) according to the manufacturer’s protocol and HTLV infection was confirmed by RFLP (22).”

2. The data do not support conclusions about susceptibiltity. Susceptibility to viral infections is mediated by many factors and this small study with low numbers of HPgV+ cases measured at one time point retrospectively cannot provide any data about susceptibility.

RESPONSE: The authors agree with the referee's observation and the conclusion has been changed, leaving only what refers to the prevalence of HPgV-1. The text was corrected and was modified as bellow: 

Abstract (lines 52-54): “Conclusion: This study indicated that the prevalence of HPgV-1 infection is low in HTLV-1 carriers in Belém, Pará, and probably does not alter the clinical course of HTLV-1 infection, however, further studies are still needed.”

Lines 250-256: “In conclusion, the prevalence of HPgV-1 infection in patients with HTLV-1 in the city of Belém is relatively similar to the prevalence found in healthy individuals (seronegative HTLV-1), however this may be influenced by the small sample size. Thus, further studies are needed to confirm that HTLV-1 and HPgV-1 coinfection does not affect the course or development of symptoms related to HTLV-1 infection in affected individuals, since the pathological mechanisms of these viruses are quite different.”

3. The age difference between the HTLV+ group and the control group is an issue, as younger subjects may be more likely to harbor HPgV as they have had less time to clear it. An age matched control group may show different results.

RESPONSE: We appreciate the observation. As the control group used in the study were blood donors who were also used in other studies of the Virology Laboratory, it was not possible to match the case and control by the age of the participant, considering the main objective of the study in verifying the prevalence of HPgV-1, especially in the population with HTLV. However, this pairing strategy will be used in the continuation of this study.

4. While the study shows no difference in clinical outcomes of HTLV based on HPgV status, the numbers are too small to make any conclusions about this. However no obvious safety signal is seen.

RESPONSE: We appreciate and agree with the observation. Therefore, we added in the study's discussion that the sample size was an important limitation of the study. In addition, we emphasize that the study, even with this limitation, is the first to describe the prevalence of HPgV-1 in HTLV-1 carriers in the north of Brazil, which will serve as a basis for future work in this area.

Lines 239-246: “This study has limitations and should be considered. The small sample size makes it difficult to assess and, consequently, the indication or safe exclusion of possible associations. Another fact to be considered is the instability of the RNA molecule that can be degraded more easily over time, so the time between the collection and the evaluation of the presence of RNA in the samples of participants with low viral load may have caused a false result negative. However, this study presented relevant information on HPgV-1 infection and is also the first investigation of its kind in this remote region of Brazil.”

Lines 249-255: “In conclusion, the prevalence of HPgV-1 infection in patients with HTLV-1 in the city of Belém is relatively similar to the prevalence found in healthy individuals (seronegative HTLV-1), however this may be influenced by the small sample size. Thus, further studies are needed to confirm that HTLV-1 and HPgV-1 coinfection does not affect the course or development of symptoms related to HTLV-1 infection in affected individuals, since the pathological mechanisms of these viruses are quite different.”

5. Grammatical issues with sentence on lines 63-65 under Introduction.

RESPONSE: Review and correction performed. The text was modified as bellow: 

Lines 63-65: “Although it has a high prevalence (studies suggest that there are ~ 750 million people with HPgV-1 infection worldwide), there is limited evidence for HPgV-1 as a primary etiological factor in human diseases [3].”

Reviewer #2: This paper describes HPgV-1 prevalence in people with (n=74) and without (n=84) HTLV-I infection living in Northern Brazil. The prevalence was 5.4% in those with HTLV-I (n= 4), and higher (9.5%) in those without HTLV-I (n=8). Due to the small number of HPgV-1 infections, these differences were not statistically significant. The following represent a number of suggestions that would strengthen the paper.

Abstract:

1. A second human pegivirus (HPgV-2) has been identified, so the HPgV described here is best called HPgV-1.

RESPONSE: We appreciate the observation and added the number 1 in all the acronyms of the HPgV.

2. The conclusion that HTLV-1 “does not change the susceptibility of patients to acquiring an HPgV infection” should be qualified, as the statistical power to suggest an increase or decrease in infection given the numbers is very low.

RESPONSE: We appreciate and agree with the observation. The authors removed from the manuscript, especially from the conclusions, the questions regarding the relationship of HTLV-1 and the susceptibility to infection by HPgV-1 due to the sample size. In addition, we added in the study limitations that the small sample size does not allow us to safely conclude anything about susceptibility. 

Lines 239-246: “This study has limitations and should be considered. The small sample size makes it difficult to assess and, consequently, the indication or safe exclusion of possible associations. Another fact to be considered is the instability of the RNA molecule that can be degraded more easily over time, so the time between the collection and the evaluation of the presence of RNA in the samples of participants with low viral load may have caused a false result negative. However, this study presented relevant information on HPgV-1 infection and is also the first investigation of its kind in this remote region of Brazil.”

Introduction

3. In addition to surrogate markers of HIV-1 disease progression mentioned in lines 75 – 79, the association between HPgV-1 and survival in HIV-1 infected people is not mentioned. The data are extensive showing reduced mortality in HIV-1 patients, even in incident infection (Vahidnia et al, JID, 2012). Also, survival in Ebola-infected people was higher in those with HPgV-1 infecton (Lauck et al., J Virol, 2014). This information should be noted.

RESPONSE: We appreciate the observation and add excerpts in the introduction of the manuscript that highlights the results of the scientific studies suggested by the referee. The paragraph was modified as below: 

Lines 80-84: “HPgV-1, which can help in the long life of those infected with HIV-1 [7,11,12]. Still intriguing, a study also indicated that HPgV-1 can interact with the host's immune system and modulate the super-exuberant immune response of the pathogenesis related to Ebola virus (EBOV) infection [13].”

Methods: No comments

Results:

4. The question of HPgV-1 prevalence and age differentiation is not possible to address due to the small number of individuals studied in age groups < 30. It appears there were 7 in the HTLV-1-positive group, and it is not clear how many individuals were in this age group in the HPgV-1+HTLV-1 coinfection group (only data for the 4 HPgV-1 cases were provided, no denominator).

RESPONSE: We appreciate the observation and agree that this information was not clear. In the group of HTLV-1 carriers, only 7 individuals were under 30 years of age, as shown in table 2, and none of them had co-infection with HPgV-1. We modified table 2, putting the number 0 (zero) in place of the hyphen for better understanding.

In conclusion, this is the second study to examine HPgV-1 infection in HTLV-I infected adults, and the first from this geographical location. Although the data do not identify significant differences in HPgV-1 prevalence by HTLV-1 status, age, sex, etc., the power to detect differences is limited due to the small number of HPgV-1 infections (n=12/158). There are many places throughout the manuscript where attention to English usage should be provided.

RESPONSE: We appreciate the comments of the reviewer and the English language was changed accordingly.

We thank very much the referees for their time going through the manuscript. The changes proposed were important to improve the presentation of the data.

Best regards

Luiz Fernando Almeida Machado

Virology Laboratory

Federal University of Pará

---

## [Decision Letter · Decision Letter 1]

22 Apr 2020

Low prevalence of human pegivirus 1 (HPgV-1) in HTLV-1 carriers from Belém, Pará, North Region of Brazil

PONE-D-20-02565R1

Dear Dr. Machado,

We are pleased to inform you that your manuscript has been judged scientifically suitable for publication and will be formally accepted for publication once it complies with all outstanding technical requirements.

With kind regards,

Jason Blackard, PhD

Academic Editor

PLOS ONE

Additional Editor Comments (optional):

None

Reviewers' comments:

Reviewer's Responses to Questions

**Comments to the Author**

1. If the authors have adequately addressed your comments raised in a previous round of review and you feel that this manuscript is now acceptable for publication, you may indicate that here to bypass the “Comments to the Author” section, enter your conflict of interest statement in the “Confidential to Editor” section, and submit your "Accept" recommendation.

Reviewer #2: All comments have been addressed

2. Is the manuscript technically sound, and do the data support the conclusions?

Reviewer #2: Yes

3. Has the statistical analysis been performed appropriately and rigorously? 

Reviewer #2: Yes

4. Have the authors made all data underlying the findings in their manuscript fully available?

Reviewer #2: Yes

5. Is the manuscript presented in an intelligible fashion and written in standard English?

Reviewer #2: Yes

6. Review Comments to the Author

Reviewer #2: The revised manuscript addresses my concerns in all cases when possible, and noted the limitations when not possible.

7. PLOS authors have the option to publish the peer review history of their article (what does this mean?). If published, this will include your full peer review and any attached files.

Reviewer #2: No

---

## [Editor Report · Acceptance letter]

24 Apr 2020

PONE-D-20-02565R1 

Low prevalence of human pegivirus 1 (HPgV-1) in HTLV-1 carriers from Belém, Pará, North Region of Brazil 

Dear Dr. Machado:

I am pleased to inform you that your manuscript has been deemed suitable for publication in PLOS ONE. Congratulations! Your manuscript is now with our production department. 

With kind regards,

on behalf of

Dr. Jason Blackard 

Academic Editor

PLOS ONE